# Women's self-help group participation and discussion of reproductive coercion: Associations with past experiences of violence among family planning clients in Kenya

Jasmine Uysal[1]*, Nicole E. Johns[1], Chi-Chi Undie[2], Wilson Liambila[2], Erin Pearson[1], Elizabeth Reed[3], Jay G. Silverman[1,4]

**1** Division of Infectious Disease and Global Public Health, Department of Medicine, Center on Gender Equity and Health, University of California San Diego, La Jolla, California, United States of America, **2** Population Council, Nairobi, Kenya, **3** School of Public Health, San Diego State University, San Diego, California, United States of America, **4** Center for Community-Engaged Research on Gender-Based Violence, Celia Scott Weatherhead School of Public Health and Tropical Medicine, Tulane University, New Orleans, Louisiana, United States of America

\* juysal@health.ucsd.edu

## Abstract

Reproductive coercion (RC) and intimate partner violence (IPV) are prevalent in Kenya and undermine women's reproductive agency; community-based women's groups may offer an opportunity to address these issues. We quantitatively examined women's participation in women-only self-help groups, described discussions of RC within these groups, and assessed differences based on prior experiences of RC and IPV among a clinic-based sample of Kenyan women seeking family planning services to inform future programming. Data were collected from 659 women of reproductive age seeking family planning services at six private clinics in Nairobi as part of the baseline for an intervention to address RC and IPV within family planning counseling. We used descriptive statistics, bivariate hypothesis testing, and adjusted mixed-effect logistic regression models to examine the relationship between self-help group participation and lifetime experience of RC and IPV. Using the same methods, we analyzed group discussions about RC in relation to lifetime experiences of abuse among women who had recently participated in groups. Additionally, we explored associations between RC discussions and the type of self-help group (e.g., economic, charitable) using bivariate tests. Over half of women reported prior participation in women-only self-help groups, with most participating in economic groups (63%). More than one in two women reported prior IPV and one in three reported prior RC. Self-help group participation was not significantly associated with prior experiences of RC or IPV. However, women who had ever experienced RC were more likely to report discussions about RC within groups (AOR 1.82 95% CI 1.06-3.14). Discussions of RC were less common in economic groups. Women-only self-help group

**Data availability statement:** Deidentified data are available on the Dryad repository DOI: https://doi.org/10.5061/dryad.9zw3r22n1

**Funding:** The author(s) received no specific funding for this work.

**Competing interests:** The authors have declared that no competing interests exist.

participation is common and acceptable among Kenyan women, including those experiencing violence. However, economic groups—though widespread—discuss RC less often, indicating a key opportunity to integrate RC/IPV programming into these settings to strengthen community-based support.

## Introduction

Reproductive coercion (RC) is a form of gender-based violence involving male partner or family member behaviors that reduce control of women and girls over contraceptive use and pregnancy decisions [1,2]. Between 3% and 35% of reproductive aged women in need of contraception report prior RC, depending on the setting and population; generally higher rates are found among women actively seeking contraceptive care [3–6]. RC can include pregnancy promoting behaviors such as contraceptive sabotage which includes hiding, destroying, or inhibiting access to contraceptive methods or pregnancy coercion which includes behaviors such as pressuring or forcing women to become pregnant or to carry a pregnancy to term. RC may also include pregnancy preventing behaviors such as forcing or pressuring women to use a contraceptive method or have an abortion against their wishes [1]. RC has been shown to be closely linked to intimate partner violence (IPV) across global settings [2,7,8] and women reporting both RC and IPV are at increased risk of experiencing negative reproductive health outcomes including unintended pregnancy [9–11].

In Kenya the estimated prevalence of lifetime and past year IPV is higher than the global average [12] and RC is prevalent [5]. Estimates from the 2022 Kenya Demographic and Health Surveys indicate that 34% of women have experienced IPV [13] and population-based surveys among married and cohabiting women in Kenya from 2021 show that 7% have experienced RC in the past year alone [6]. In a recent study among women seeking contraceptive clinical care in Nairobi, 80% of IPV survivors were found to have experienced prior RC [14]. One qualitative study among female contraceptive clients in Nairobi described the severe forms of pregnancy-promoting RC that women in Kenya experience including contraceptive sabotage and pregnancy coercion, and the intersection of these experiences with IPV, including violence and threats to become pregnant or to stop using family planning. Women described how they coped with RC by using contraceptive methods covertly and intentionally hiding their use from their male partner or family members who desired that they become pregnant [15]. The findings from Kenya are not unique, studies now document the prevalent and severe effects of RC and the importance of covert contraceptive use across various global contexts [1–3,6,8,11,16].

Studies have also demonstrated the importance of female peer-support for women's ability to cope with experiences of gender-based violence [17] and that lack of social support is associated with greater risk for IPV and reduced ability to leave violent relationships [18,19]. Qualitative research from Kenya has found that women reporting RC commonly rely on peer-support to cope with these experiences, including sharing information and applying coping strategies to maintain reproductive

control in the face of male partner or family opposition to contraceptives such as how to use a method without a male partner knowing [15]. In this study, women went so far as to ask other women in their communities to get family planning for them or to keep family planning at their friends' homes so their partner would not find out. Women also reported that financial savings clubs specifically provided an important source of information and support around family planning use, particularly in the face of opposition.

Women-only self-help groups are one existing community structure where women and girls have been shown to find social support [20], increase awareness about their rights to be in a safe and healthy relationship, become economically self-sufficient, and increase control over reproductive and family planning decisions [21–23]. Women's economic self-help groups including informal savings and loan groups (common types in Kenya include *Chamas or* Village Savings and Loan Associations) are particularly widespread across communities in sub-Saharan Africa [24] and in other regions globally, such as South Asia. Beyond social support, economic self-help groups also offer women opportunities to increase their own economic self-sufficiency which studies have shown may afford them more power in intimate relationships, agency to leave abusive relationships, and agency to seek and obtain contraceptives [25–28]. *Chamas* are an essential part of the Kenya Ministry of Health's community health strategy [29] as participation in these groups has been associated with improved maternal and child health outcomes in this context [30].

Despite the promise of women-only self-help groups to provide social support and help women cope with gender-based violence, including RC and IPV, to date, there is no quantitative evidence on whether participation in such groups is associated with experiences of RC and IPV or whether RC is actively discussed in these settings. To inform future programming, this study uses cross-sectional data to explore whether women's self-help groups may already serve as informal spaces of support for women experiencing RC and IPV, and whether those prior experiences of RC or IPV influence participation or discussion within the groups. Specifically, we assess the relationship between self-help group participation, discussion of RC within groups, and women's prior experiences of RC and IPV, among a sample of women of reproductive age seeking contraceptive services at private, community-based clinics in Nairobi, Kenya. We also explore whether the likelihood of RC-related discussion varies by type of self-help group, with particular attention to economic self-help groups.

## Methods

### Ethics statement

This parent study was approved by the University of California San Diego's Human Research Protections Program (Protocol 170084, February 7, 2017), the Population Council Institutional Review Board (Protocol 797, January 18, 2017) and the Kenyatta National Hospital-University of Nairobi Ethics and Research Committee (Protocol P945/12/2016, March 2, 2017). All participants completed written informed consent. A waiver of parental consent was obtained for minors aged 15–17 years old based on the conditions that research presents no greater than minimal risk to children and parental consent is not a reasonable requirement to protect safety of participants who may be accessing family planning or services for violence support without parental knowledge. The secondary analysis for which this study was based was completed entirely on secondary de-identified data with no interaction with participants and thus does not qualify as human subjects research.

### Data source

This study utilizes secondary, cross-sectional data from a convenience sample of women seeking family planning services from six private clinics in Nairobi, Kenya collected between July 2018 and December 2018. Data were drawn from the baseline survey of the ARCHES (Addressing Reproductive Coercion in Health Settings) Kenya adaptation and evaluation study. ARCHES is a brief intervention delivered within routine family planning services to increase women's ability to

cope with experiences of IPV and RC to support women's reproductive agency. The six private clinics were all operated by Family Health Options of Kenya (FHOK), the local implementing partner for the parent evaluation study. Clinics were selected based on programmatic priorities, geographic location in or near Nairobi, and feasibility of intervention delivery; the resulting sample is not intended to be representative of the general population of reproductive-age women in Kenya or Nairobi. Further detail on the study design, including power calculations, are described in the parent study protocol [31].

Women aged 15–49 years old attending study clinics who reported interest in receiving family planning services at their appointment that day and not being currently pregnant or sterilized, were able to communicate in English or Kiswahili, were planning to remain in the area for a period of six months (duration of the study) and had a phone that would allow them to be re-contacted for follow-up surveys were eligible for participation. Those who were interested completed the baseline survey immediately prior to their appointment.

## Measures

We examined two aspects of self-help group involvement, women's self-help group participation and discussion of RC topics within self-help groups. Self-help group participation was asked directly: "Do you attend any group meetings or activities that are for women only?" Among those who responded 'yes', they were asked how many times they attended this group in the prior three months. Women who indicated participation in a group were also asked the type of group they attend most frequently, with answer options: religious group, charitable/social work, sports/hobbies, loans/savings clubs, or other. Discussion of RC topics within self-help groups was assessed among women who responded that "yes" they attended group meetings or activities for women only. To assess discussion on RC within groups, participants were asked whether any of the following topics were discussed in any of the group meetings they attended within the past three months: 1) Male partners or in-laws that pressure or force women to stop using family planning (family planning related RC), 2) Male partners or in-laws that pressure or force women to become pregnant (pregnancy related RC), 3) How to use family planning without a partner or other family member knowing (covert family planning use). Responses were binary (yes/no). Women who indicated 'don't know' were coded as 'no' for analyses (n selecting 'don't know' < 1%). We examined these items separately and as a combined 'participated in any of the three discussion topics' (yes/no) variable.

Two independent variables, ever and past-year experience of RC victimization and IPV victimization, were used in this analysis. RC was evaluated based on 11 items capturing contraceptive sabotage and pregnancy coercion [31]. RC was indicated if the respondent indicated 'yes' to any item. Women who responded yes to any RC item were also asked when the last time a partner did any of these things to them; we used this to determine past year experience. IPV was assessed as experience of physical or sexual violence, including whether a male partner had ever: "pushed you, shaken you, or thrown something at you," "slapped you," "twisted your arm or pulled your hair," "hit you with his fist or with something that could hurt you," "kicked you, dragged you, or beat you up," "choked you or tried to burn you," or "forced you to have sex or do something sexual with you when you didn't want to." IPV was indicated if the respondent indicated 'yes' to any item. As with RC, women reporting IPV were asked about the timing of their most recent experience to assess past-year exposure. Women missing information on RC or IPV items were excluded from the respective analyses.

Covariates included women's age (years), education (primary or less, secondary, higher than secondary), literacy (yes/no), marital status (married or cohabiting with a partner, not married), parity (0, 1, 2+), past year employment (yes/no), and past 30 days food insecurity (yes/no).

## Statistical analyses

We first present descriptive statistics regarding demographics and outcomes by women's self-help group participation ever and within the past three months. Pearson's Chi-square tests were used to evaluate differences in participation for categorical and binary measures, and t-tests (assuming unequal variances) were used for continuous measures. We

present descriptive statistics regarding frequency of participation, group type, and discussion within groups. Next, we use Chi-squared tests for unadjusted bivariate comparisons and mixed-effects logistic regression models (both unadjusted and adjusted) to assess associations between women-only self-help group participation and outcomes (RC and IPV, ever and past year) in the full sample. We then repeated these analyses with any discussion of RC in groups in thepast three monthsas the outcome of interest, limited to women who reported participating in self-help groups during that period. These analyses were conducted for discussion of any RC topic, as well as for each of the three specific topics. Adjusted models accounted for clinic-level similarities via clinic random intercept specifications, and controlled for age, education, literacy, marital status, parity, employment, and food security via fixed effects. Finally, we assessed discussion of RC (any and specific topic) by women's self-help group type, using bivariate chi-squared test comparisons; due to small numbers of sports/hobbies group participants (n = 3) and 'other' type group participants (n = 7), only religious groups, charitable/ social work, and loans/savings clubs were included in these comparisons. All analyses were completed on non-missing data ; no imputation was performed.

Significance was set at p < 0.05 for all comparisons and marginal significance at p < 0.1; odds ratios (ORs) and adjusted odds ratios (AORs), along with 95 percent confidence intervals (CIs) are reported for regression results. All analyses were conducted using STATA 15.1.

## Results

In total, 659 women provided baseline survey response and were included in the present analysis. More than one in three women reported ever experiencing RC (36.9%), and one in seven (14.7%) reported past year RC (Table 1). More than half of women reported ever experiencing IPV (55.2%), one in four (27.2%) in the past year . Half (50.1%) of women reported ever participating in a women's group, and most of those who had participated attended one within the past three months (92.1% of ever attendees, 46.1% of women overall; Tables 1 and 2). Loans and savings clubs were the most reported type of women's group attended (63.3%), followed by charitable work/social work (17.3%) and religious groups (16.4%). Women's group participation was significantly more common among women who were older, married, employed, and who had one or more children; this was true for both ever and past three-month participation.

Among past three-month women's group participants (N = 304), 44.7% reported discussing at least one of the three specified topics related to reproductive coercion (family planning related RC, pregnancy related RC, and/or covert family planning use; Table 2). Discussion of each specific topic related to RC was reported by approximately one third of past three-month group participants: 27.6% reported family planning RC discussion, 31.6% reported pregnancy RC discussion, and 32.2% reported covert family planning use discussion.

There were no significant differences in reports of RC ever, IPV ever, or IPV past year by women's group participation (Table 3). Women who experienced RC in the past year were marginally less likely to participate in a women's group in unadjusted comparisons (42.3%, vs 51.7%, p = 0.09); in adjusted comparisons, however, there was no significant association between past year RC and women's group participation (AOR 0.72, 95% CI 0.44-1.16).

In models limited to past three-month women's group participants, women who had ever experienced RC were more likely than women who had never experienced RC to report discussion of at least one of three examined topics at a women's group (56.5% vs 38.3%, p = 0.002; Table 4). This relationship remained significant in both unadjusted and adjusted models (OR 1.82, 95% CI 1.10-3.02; AOR 1.82, 95% CI 1.06-3.14). No significant differences in RC discussion were observed by whether women reported past year RC; sample size for these comparisons was limited as only n = 37 women both attended a women's group in the prior three months and reported RC in the prior year. No differences in discussion were observed for women reporting *ever* experience of IPV, however, women who experienced IPV in the prior year were marginally more likely than those who did not experience past year IPV to report discussion of at least one RC-related topic (54.7% vs 41.4%, p = 0.045). This relationship, however, attenuated to nonsignificance in adjusted models (AOR 1.38, 95% CI 0.75-2.54).

**Table 1. Sample sociodemographic characteristics, total sample and by women's group participation ever and past 3 months.**

| Variable | Total n (%) | Women's Group Participation Ever | | | Women's Group Participation Past 3 Months | | |
|---|---|---|---|---|---|---|---|
| | | Yes | No | p-value* | Yes | No | p-value* |
| N (%) | 659 (100) | 330 (50.1) | 329 (49.9) | | 304 (46.1) | 355 (53.9) | |
| Age (M±SD) | 27.3±7.3 | 29.3±8.08 | 25.3±5.7 | p<0.001 | 29.4±8.1 | 25.5±5.9 | p<0.001 |
| Education | | | | p=0.58 | | | p=0.23 |
| Primary or less | 145 (22.0) | 68 (20.6) | 77 (23.4) | | 64 (21.1) | 81 (22.8) | |
| Secondary | 238 (36.1) | 118 (35.8) | 120 (36.5) | | 102 (33.6) | 136 (38.3) | |
| Tertiary or higher | 276 (41.9) | 144 (43.6) | 132 (40.1) | | 138 (45.4) | 138 (38.9) | |
| Literacy | | | | p=0.29 | | | p=0.49 |
| Yes | 636 (96.5) | 321 (91.3) | 315 (95.7) | | 295 (97.0) | 341 (96.1) | |
| No | 23 (3.5) | 9 (2.7) | 14 (4.3) | | 9 (3.0) | 14 (3.9) | |
| Marital Status | | | | p=0.03 | | | p=0.03 |
| Married | 429 (65.1) | 228 (69.1) | 201 (61.1) | | 211 (69.4) | 218 (61.4) | |
| Not Married | 230 (34.9) | 102 (30.9) | 128 (38.9) | | 93 (30.6) | 137 (38.6) | |
| Parity | | | | p<0.001 | | | p<0.001 |
| Nulliparous | 147 (22.3) | 51 (15.5) | 96 (29.2) | | 49 (16.1) | 98 (27.6) | |
| Uniparous | 201 (30.5) | 96 (29.1) | 105 (31.9) | | 89 (29.3) | 112 (31.5) | |
| Multiparous | 311 (47.2) | 183 (55.5) | 128 (38.9) | | 166 (54.6) | 145 (40.8) | |
| Employed past 12mo | | | | p<0.001 | | | p<0.001 |
| Yes | 455 (69.0) | 261 (79.1) | 194 (59.0) | | 243 (79.9) | 212 (59.7) | |
| No | 204 (31.0) | 69 (20.9) | 135 (41.0) | | 61 (20.1) | 143 (40.3) | |
| Food Insecurity | | | | p=0.30 | | | p=0.28 |
| Yes | 120 (18.2) | 55 (16.7) | 65 (19.8) | | 50 (16.4) | 70 (19.7) | |
| No | 539 (81.8) | 275 (83.3) | 264 (80.2) | | 254 (83.6) | 285 (80.3) | |
| Any Reproductive Coercion Victimization Ever | | | | p=0.51 | | | p=0.47 |
| Yes | 243 (36.9) | 118 (35.8) | 125 (38.0) | | 108 (35.5) | 135 (38.0) | |
| No | 414 (62.8) | 212 (64.2) | 202 (61.4) | | 196 (64.5) | 218 (61.4) | |
| Missing | 2 (0.3) | 0 (0.0) | 2 (0.6) | | 0 (0.0) | 2 (0.6) | |
| RC Victimization Past Year | | | | p=0.09 | | | p=0.08 |
| Yes | 97 (14.7) | 41 (12.4) | 56 (17.0) | | 37 (12.2) | 60 (16.9) | |
| No | 559 (84.8) | 289 (87.6) | 270 (82.1) | | 267 (87.8) | 292 (82.3) | |
| Missing | 3 (0.5) | 0 (0.0) | 3 (0.9) | | 0 (0.0) | 3 (0.8) | |
| Any Intimate Partner Violence Victimization Ever | | | | p=0.48 | | | p=0.21 |
| Yes | 364 (55.2) | 187 (56.7) | 177 (53.8) | | 176 (57.9) | 188 (53.0) | |
| No | 290 (44.0) | 141 (42.7) | 149 (45.3) | | 126 (41.4) | 164 (46.2) | |
| Missing | 5 (0.8) | 2 (0.6) | 3 (0.9) | | 2 (0.7) | 3 (0.8) | |
| IPV Victimization Past Year | | | | p=0.12 | | | p=0.18 |
| Yes | 179 (27.2) | 81 (24.5) | 98 (29.8) | | 75 (24.7) | 104 (29.3) | |
| No | 475 (72.1) | 247 (74.8) | 228 (69.3) | | 227 (74.7) | 248 (69.9) | |
| Missing | 5 (0.8) | 2 (0.6) | 3 (0.9) | | 2 (0.7) | 3 (0.8) | |

*Note. Chi-square or t-tests, where appropriate, were used to detect differences in variables by women's group participation.

**Table 2. Women's Group Participation Frequency, by Sample of Women Ever Attending Groups (n = 330).**

| Variable | n (%) |
|---|---|
| ***Total ever participation*** | ***330 (100)*** |
| Frequency of Attendance Past 3 Months | |
| Not at all | 26 (7.9) |
| One to two times | 78 (23.6) |
| Three to five times | 117 (35.5) |
| More than five times | 109 (33.0) |
| Any Participation in Past 3 Months | |
| Yes | 304 (92.1) |
| No | 26 (7.9) |
| Women's Group Type | |
| Religious group | 54 (16.4) |
| Charitable work/Social work | 57 (17.3) |
| Sports/hobbies | 3 (0.9) |
| Loans and savings clubs | 209 (63.3) |
| Other | 7 (2.1) |
| ***Total past 3 month participation*** | ***304 (100)*** |
| Women's Group RC Discussion (any of below 3 topics) Past 3 Months | |
| Yes | 136 (44.7) |
| No | 168 (55.3) |
| Discussion Topic 1 Past 3 Months: Male partners or in-laws that pressure or force women to stop using family planning | |
| Yes | 84 (27.6) |
| No | 219 (72.0) |
| Don't know | 1 (0.3) |
| Discussion Topic 2 Past 3 Months: Male partners or in-laws that pressure or force women to become pregnant | |
| Yes | 96 (31.6) |
| No | 207 (68.1) |
| Don't know | 1 (0.3) |
| Discussion Topic 3 Past 3 months: How to use family planning without a partner or other family member knowing | |
| Yes | 98 (32.2) |
| No | 206 (67.8) |

Examined separately by discussion topic, we found consistent positive associations between ever experiencing RC and discussion family planning and pregnancy related RC and covert family planning use (Additional files 1, 2, 3). Associations between individual discussion topics and ever experiencing RC were strongest for discussion of covert family planning use (OR 2.21, 95% CI 1.31-3.73; AOR 2.10, 95% CI 1.20-3.67). Additionally, unadjusted bivariate comparisons and unadjusted regression models indicate significant associations between discussion of covert family planning use and both past year RC and past year IPV (RC past year OR 2.06, 95% CI 1.01-4.20; IPV past year OR 1.87, 95% CI 1.07-3.27). These associations were not significant in adjusted comparisons, however (RC past year AOR 1.44, 95% CI 0.66-3.14; IPV past year 1.59, 95% CI 0.85-2.97).

**Table 3. Unadjusted and Adjusted Logistic Regression Models to Assess the Association Between Reproductive Coercion and Intimate Partner Violence Victimization and Women's Group Participation Ever, Total Sample (N = 659).**

| Variable | Percent participating in Women's Groups | Chi-squared p-value | OR (95% CI) | AOR (95% CI) |
|---|---|---|---|---|
| Reproductive Coercion Victimization Ever | | 0.51 | | |
| Yes | 48.6% | | 0.90 (0.64-1.27) | 0.83 (0.58-1.21) |
| No (ref) | 51.2% | | | |
| Reproductive Coercion Victimization Past Year | | 0.09 | | |
| Yes | 42.3% | | 0.66 (0.42-1.04)^ | 0.72 (0.44-1.16) |
| No (ref) | 51.7% | | | |
| Intimate Partner Violence Victimization Ever | | 0.48 | | |
| Yes | 51.4% | | 1.25 (0.90-1.73) | 1.10 (0.77-1.58) |
| No (ref) | 48.6% | | | |
| Intimate Partner Violence Victimization Past Year | | 0.12 | | |
| Yes | 45.3% | | 0.80 (0.56-1.14) | 0.85 (0.57-1.28) |
| No (ref) | 52.0% | | | |

Note. Logistic regression models were adjusted by the following demographic variables: age, education, literacy, marriage status, parity, employment, and food security.

^$p < .1$

*$p < .05$

**$p < .01$

***$p < .001$

**Table 4. Unadjusted and Adjusted Logistic Regression Models to Assess the Association Between Reproductive Coercion and Intimate Partner Violence Victimization and Past 3 months Women's Group Discussion, among those who attended a Women's Group in the prior 3 months (N = 304).**

| Variable | Percent participating in Discussion | Chi-squared p-value | OR (95% CI) | AOR (95% CI) |
|---|---|---|---|---|
| Reproductive Coercion Victimization Ever | | 0.002 | | |
| Yes | 56.5% | | 1.82 (1.10-3.02)* | 1.82 (1.06-3.14)* |
| No (ref) | 38.3% | | | |
| Reproductive Coercion Victimization Past Year | | 0.22 | | |
| Yes | 54.1% | | 1.34 (0.66-2.76) | 0.98 (0.45-2.13) |
| No (ref) | 43.4% | | | |
| Intimate Partner Violence Victimization Ever | | 0.59 | | |
| Yes | 46.0% | | 1.04 (0.64-1.69) | 1.07 (0.63-1.81) |
| No (ref) | 42.9% | | | |
| Intimate Partner Violence Victimization Past Year | | 0.045 | | |
| Yes | 54.7% | | 1.54 (0.89-2.66) | 1.38 (0.75-2.54) |
| No (ref) | 41.4% | | | |

Note. Logistic regression models were adjusted by the following demographic variables: age, education, literacy, marriage status, parity, employment, and food security.

^$p < .1$

*$p < .05$

**$p < .01$

***$p < .001$

In descriptive analyses examining RC-related discussion frequencies by women's group type, we find that women attending charitable work/social work type groups were most likely to report discussion of at least one topic, and of each specific topic (70% report discussing any topic, 42% family planning RC, 42% pregnancy RC, and 60% covert family planning use) (Table 5). Loans and savings club participants had the lowest reported rates of discussion (38% report discussing any topic, 23% family planning RC, 29% pregnancy RC, and 25% covert family planning use); religious group discussion rates were similar to loans and savings club discussion rates. Differences in discussion by group type were statistically significant for any RC-related discussion (p < 0.001), and for family planning RC (p = 0.02) and covert family planning use (p < 0.001), but not for pregnancy RC (p = 0.19).

## Discussion

To our knowledge, this is the first study to assess how women's experience of RC and IPV is associated with their participation women's only in self-help groups and discussion of RC within these groups. We found that participation in self-help groups was common in this sample, with one in two women reporting participation, most frequently in economic self-help groups. Women with a history of RC or IPV were more likely to discuss RC within groups. Additionally, we found that neither IPV nor RC were associated with women's participation in self-help groups and that women may discuss RC more or less frequently depending on the type of self-help group they reported participating in. We also found that the covert contraceptive use was the most commonly discussed topic related to RC within this setting. Findings shed light on possible opportunities for intervention around RC within the context of women-only self-help groups.

We found evidence that women with a history of RC or IPV were more likely to discuss RC and covert use topics within these groups. This suggests that women's groups may act as an important resource and forum within their communities for survivors to engage in dialogue around RC, share experiences, and seek support to maintain their reproductive agency in the face of active opposition. Prior qualitative research among women seeking family planning services in Kenya supports this assumption, finding that female peer-support networks were common and essential for coping with experiences of RC and IPV, allowing women to use methods covertly when their partner was actively opposing their use, and to even support leaving violent relationships [15]. Additionally, findings from the intervention evaluation from which these data were analyzed support that formal survivor services are inaccessible and unacceptable to many women [32].

**Table 5. Discussion frequencies by group type, among those participating in a women's group in the prior 3 months (n = 295)ᵃ.**

| Variable | Religious group | Charitable work/ social work | Loans and savings clubs | Chi-Squared P-value |
|---|---|---|---|---|
| Any discussion | | | | <0.001 |
| Yes | 22 (43.1) | 35 (70.0) | 73 (37.6) | |
| No | 29 (56.9) | 15 (30.0) | 121 (62.4) | |
| Male partners or in-laws that pressure or force women to stop using family planning | | | | 0.02 |
| Yes | 12 (23.5) | 21 (42.0) | 45 (23.2) | |
| No | 39 (76.5) | 29 (58.0) | 149 (76.8) | |
| Male partners or in-laws that pressure or force women to become pregnant | | | | 0.19 |
| Yes | 15 (29.4) | 21 (42.0) | 56 (28.9) | |
| No | 36 (70.6) | 29 (58.0) | 138 (71.1) | |
| How to use family planning without a partner or other family member knowing | | | | <0.001 |
| Yes | 17 (33.3) | 30 (60.0) | 48 (24.7) | |
| No | 34 (66.7) | 20 (40.0) | 146 (75.3) | |

ᵃWomen participating in sports/hobbies groups in the past 3 months (n = 3) and women participating in 'other' groups in the past 3 months (n = 6) excluded due to small cell sizes.

In fact, when asked on how to improve the intervention, the most common recommendation from providers and patients in this study was to develop a community-based model to build peer-support for RC and IPV in addition to clinic-based services [forthcoming]. Together, these findings suggest that women's self-help groups may offer opportunities to engage and assist women who have experienced gender-based violence, including support for and coping with RC experiences, specifically.

Prior studies in South Asia and Sub-Saharan Africa have documented that women experiencing IPV limited freedom of movement outside the home and may be less likely to seek certain types of health care (such as antenatal care) [33,34]. Interestingly, we found that, among Kenyan women in this sample seeking contraceptive methods, women experiencing RC or IPV were no less likely to participate in self-help groups, including economic groups, than women not reporting prior victimization. These findings could be due to the nature of this sample of women, who may have increased agency over typical survivors to leave the home to seek contraceptive care, or because freedom of movement is less restricted for women in the Kenyan context, even those experiencing violence. In either case, our findings support that intervention for RC and IPV within self-help groups could be a viable venue to reach many women in need of support for violence experiences. Additionally, as women's-only self-help groups often reside within normative community structures (economic groups, church groups, etc.) these venues may have the potential to reach women in need of violence support who are unable or hesitant to seek formal healthcare or contraceptive services, a hypothesis which should be explored in future studies.

When examined separately by women's group type, we found that RC and covert contraceptive use discussion topics were most frequently reported within charitable work/social work groups and not economic groups, highlighting a potential opportunity for further engagement in economic empowerment women-only self-help groups. Studies have documented that economic reliance on a male partner can increase women's susceptibility to violence [35,36], making them unable to leave abusive relationships and even purchase a family planning method [37], supporting the need for joint interventions to increase economic and reproductive empowerment. Economic empowerment itself has also shown to be protective against experience of certain forms of IPV within Sub-Saharan Africa [38]. Economic self-help groups were the most prevalent form of group participation reported in this sample and are the predominant model in many communities, particularly in Kenya and Sub-Saharan Africa (e.g., Chamas) [24,29,30]. Prior studies have shown that these groups can improve family planning and economic outcomes [20–22,24,30,39]. As such, they may offer a feasible, sustainable, and scalable platform for interventions aimed at increasing awareness and support of RC and IPV and promoting economic empowerment [20]. Given our finding that RC discussion was less frequent in economic self-help groups, there may be an unfulfilled need to further engage these groups.

This study has several limitations. As a cross-sectional study based on secondary, retrospective, self-report data (with no follow-up validation of participant reports), our findings cannot be inferred as causal. Additionally, given that we utilized secondary data from a pilot study involving a convenience sample of family planning patients visiting private facilities in Nairobi, findings cannot be generalized to the broader population of women of reproductive age in Kenya or Nairobi, nor can they be said to represent those violence survivors who are the most impacted and likely unable to access routine reproductive health services. We are somewhat limited by our measurement and power; some outcomes may have been insignificant due to insufficient power rather than true non-effects and our binary measurement of RC discussion is not able to tease out the differentiations on the quality of conversations nor how topics were discussed within groups (broadly or in private conversations). These aforementioned limitations have important implications for the acceptability of discussing these sensitive topics in a group setting, for which follow-up studies are required. Finally, it is likely that the nature of women-only self-help groups and women's participation varied widely, which we were unable to account for in this analysis due to lack of detailed measures on group structure or participation quality. Despite these limitations, this study provides a valuable glimpse into the potential relationships between self-help groups and Kenyan women's experiences with RC and IPV to inform future research and intervention.

## Conclusions

Our findings highlight the potential of women-only self-help groups as a valuable platform to support women experiencing reproductive coercion (RC) and intimate partner violence (IPV). In this first quantitative study of women's participation in self-help groups and discussions of RC, we found that both participation and discussions of RC were common, and neither RC nor IPV inhibited group involvement. These findings demonstrate the social acceptability and widespread use of self-help groups among Kenyan women, including those who have experienced gender-based violence, and underscore their potential to provide critical support for survivors. However, economic self-help groups, while the most prevalent, had lower rates of RC discussions, suggesting an important opportunity to integrate RC and IPV programming into these groups. Such integration could expand community-based support for women facing challenges in exercising reproductive autonomy, particularly in Kenya and other regions of Sub-Saharan Africa where economic self-help groups are widespread. These insights can guide the development of scalable, sustainable interventions to increase awareness and provide support for survivors of RC and IPV.

## Acknowledgments

We would like to thank Melanie Maino Vieytes for her contributions to the literature review informing this work.

## Author contributions

**Conceptualization:** Jasmine Uysal.

**Data curation:** Jasmine Uysal, Chi-Chi Undie, Wilson Liambila.

**Formal analysis:** Jasmine Uysal, Nicole E. Johns, Erin Pearson.

**Investigation:** Jasmine Uysal, Jay G. Silverman.

**Methodology:** Jasmine Uysal, Erin Pearson, Elizabeth Reed, Jay G. Silverman.

**Project administration:** Jay G. Silverman.

**Supervision:** Jay G. Silverman.

**Validation:** Chi-Chi Undie.

**Writing – original draft:** Jasmine Uysal, Nicole E. Johns.

**Writing – review & editing:** Chi-Chi Undie, Wilson Liambila, Erin Pearson, Elizabeth Reed, Jay G. Silverman.

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
