## [Decision Letter · Decision Letter 0]

26 Apr 2025

PGPH-D-24-02696

Participation in and discussion of reproductive coercion in women-only self-help groups - associations with prior experience of reproductive coercion and intimate partner violence among a clinic-based sample of Kenyan women

Dear Dr. Uysal,

Thank you for submitting your manuscript to PLOS Global Public Health. After careful consideration, we feel that it has merit but does not fully meet PLOS Global Public Health’s publication criteria as it currently stands. Therefore, we invite you to submit a revised version of the manuscript that addresses the points raised during the review process.

Editor comments:

The manuscript offers helpful insights into relationships between female participation in self-help groups and discussions of reproductive coercion and intimate-partner violence. The implications for intervention development are also explained well. However, I agree with Reviewer 1's suggestions for edits that would strengthen the clarity of the manuscript.

We look forward to receiving your revised manuscript.

Kind regards,

Marie A. Brault, PhD

Academic Editor

Journal Requirements:

1. Please ensure that the Title in your manuscript file and the Title provided in your online submission form are the same.

Additional Editor Comments (if provided):

Reviewers' comments:

Reviewer's Responses to Questions

**Comments to the Author**

1. Does this manuscript meet PLOS Global Public Health’s publication criteria ? Is the manuscript technically sound, and do the data support the conclusions? The manuscript must describe methodologically and ethically rigorous research with conclusions that are appropriately drawn based on the data presented.

Reviewer #1: Yes

Reviewer #2: Partly

2. Has the statistical analysis been performed appropriately and rigorously?

Reviewer #1: Yes

Reviewer #2: N/A

3. Have the authors made all data underlying the findings in their manuscript fully available (please refer to the Data Availability Statement at the start of the manuscript PDF file)?

Reviewer #1: Yes

Reviewer #2: Yes

4. Is the manuscript presented in an intelligible fashion and written in standard English?

Reviewer #1: Yes

Reviewer #2: Yes

5. Review Comments to the Author

Reviewer #1: Two minor revisions:

The title seems clunky, repetitive and not especially helpful for understanding what the study is about. SUGGESTED TITLE: Experiences and discussions of reproductive coercion and intimate partner violence among participants in women-only self-help groups in Kenya: Findings from the ARCHES Baseline Survey

In the introduction, can you describe the nature of the association you examined between experiences of RC and IPV and self-help group participation? What was the temporality of relationship from the perspective of study authors? People who experience these things seek out support from self-help groups? Or participation in self-help groups leads to these experiences?

Reviewer #2: It is a much needed research, cause I am not an expert in your methodology. However, this paper needs to be reviewed by an expert in the methodology needed. xxxxxxxxxxxxxxxxxxxxxxxxxxxxxxxxxxxxxxxxxxxxx

6. PLOS authors have the option to publish the peer review history of their article (what does this mean? ). If published, this will include your full peer review and any attached files.

**Do you want your identity to be public for this peer review?** For information about this choice, including consent withdrawal, please see our Privacy Policy .

Reviewer #1: **Yes: ** Carinne Brody

Reviewer #2: No

---

## [Decision Letter · Decision Letter 1]

16 Jul 2025

PGPH-D-24-02696R1

Women’s Self-Help Group Participation and Discussion of Reproductive Coercion: Associations with Past Experiences of Violence Among Family Planning Clients in Kenya

Dear Dr. Uysal,

Thank you for submitting your manuscript to PLOS Global Public Health. After careful consideration, we feel that it has merit but does not fully meet PLOS Global Public Health’s publication criteria as it currently stands. Therefore, we invite you to submit a revised version of the manuscript that addresses the points raised during the review process.

Editor comments:

The reviewers and I found the manuscript to be improved, with just a few remaining questions to be addressed in the methods and results. Once these are addressed, the manuscript should be set.

We look forward to receiving your revised manuscript.

Kind regards,

Marie A. Brault, PhD

Academic Editor

Journal Requirements:

Additional Editor Comments (if provided):

Reviewers' comments:

Reviewer's Responses to Questions

**Comments to the Author**

1. If the authors have adequately addressed your comments raised in a previous round of review and you feel that this manuscript is now acceptable for publication, you may indicate that here to bypass the “Comments to the Author” section, enter your conflict of interest statement in the “Confidential to Editor” section, and submit your "Accept" recommendation.

Reviewer #1: All comments have been addressed

Reviewer #3: (No Response)

2. Does this manuscript meet PLOS Global Public Health’s publication criteria ? Is the manuscript technically sound, and do the data support the conclusions? The manuscript must describe methodologically and ethically rigorous research with conclusions that are appropriately drawn based on the data presented.

Reviewer #1: Yes

Reviewer #3: Yes

3. Has the statistical analysis been performed appropriately and rigorously?

Reviewer #1: Yes

Reviewer #3: Yes

4. Have the authors made all data underlying the findings in their manuscript fully available (please refer to the Data Availability Statement at the start of the manuscript PDF file)?

Reviewer #1: Yes

Reviewer #3: Yes

5. Is the manuscript presented in an intelligible fashion and written in standard English?

Reviewer #1: Yes

Reviewer #3: Yes

6. Review Comments to the Author

Reviewer #1: Looks good - thanks.

Reviewer #3: The paper is both descriptive and inferential. The investigators used descriptive statistics, bivariate hypotheses testing, and adjusted mixed effect logistic regression models to examine the relationship between self-help group participation and lifetime experience of reproductive coercion and intimate partner violence. For women who had participated in self-help groups within the past three months, they used these same methods to analyze group discussions about reproductive coercion in relation to lifetime experiences of abuse. The presentation is fairly well written from a statistical perspective.. However, there are a few issues to be clarified.

1. Why 659 subjects ?. There obviously was no formal calculation justifying this. The Investigators do admit possible limited power as a limitation. That may or may not be as the confidence intervals on Tables 3 and 4 look reasonable.

2. Also see the first column of Table 1. How representative is this sample to the same age group of women in Kenya or Nairobi?

3. Why private clinics? See the methods section. There should be some discussion of this unless this reviewer missed it.

4. The statistics are done appropriately in both the univariate and multivariate approaches. The limitations are clearly stated, a major one being that the this is a cross-sectional study based on retrospective self-report. Was there any follow up validation of any sample of the reports?

7. PLOS authors have the option to publish the peer review history of their article (what does this mean? ). If published, this will include your full peer review and any attached files.

**Do you want your identity to be public for this peer review?** For information about this choice, including consent withdrawal, please see our Privacy Policy .

Reviewer #1: **Yes: ** Carinne Brody

Reviewer #3: No

---

## [Editor Report · Decision Letter 2]

14 Aug 2025

Women’s Self-Help Group Participation and Discussion of Reproductive Coercion: Associations with Past Experiences of Violence Among Family Planning Clients in Kenya

PGPH-D-24-02696R2

Dear Uysal,

We are pleased to inform you that your manuscript 'Women’s Self-Help Group Participation and Discussion of Reproductive Coercion: Associations with Past Experiences of Violence Among Family Planning Clients in Kenya' has been provisionally accepted for publication in PLOS Global Public Health.

Best regards,

Marie A. Brault, PhD

Academic Editor